# Exploring the External Environmental Drivers of Honey Bee Colony Development

Nuno Capela [1,*], Artur Sarmento [1], Sandra Simões [1], Sara Lopes [1], Sílvia Castro [1], António Alves da Silva [1], Joana Alves [1], Yoko L. Dupont [2], Dirk C. de Graaf [3] and José Paulo Sousa [1]

1    Centre for Functional Ecology, Department of Life Sciences, Associated Laboratory TERRA, University of Coimbra, 3000-456 Coimbra, Portugal
2    Department of Ecoscience, Aarhus University, 8000 Aarhus, Denmark
3    Department of Biochemistry and Microbiology, Ghent University, Krijgslaan 281 S2, B-9000 Ghent, Belgium
*    Correspondence: nuno.capela@uc.pt

**Abstract:** Honey bees play an important role in agricultural landscapes by providing pollination services. Throughout the season, colonies increase their population and collect resources from the available flowering plants. Besides internal mechanisms, such as the amount of brood or the availability of bees to perform foraging flights, colonies are also influenced by the climate and the surrounding landscape. Therefore, exposure to different environmental contexts leads to distinct development rates. In this study, we show how colonies develop under three different landscape contexts and explore which external variables (mostly climate and resources availability) influence the colonies' development. We installed three apiaries in three different landscapes in the Iberian Peninsula, with temporal and spatial variation in climatic conditions and resource availability. The availability of resources and their use, as well as the development of colonies throughout the season, were thoroughly investigated. These data were used to take the first step into creating an ecologically relevant landscape by calculating the number of available resources in the landscape at different points in time, based on plants' beekeeping interest as well as nectar and production. Furthermore, climatic variables were transformed into the amount of available foraging minutes that bees had to collect resources, and a theoretical threshold of optimal vs. sub-optimal conditions was also explored. Interestingly, the main drivers of colony development (measured by daily weight increase) were not the same in the tested apiaries, evidencing how colonies are indeed intrinsically connected with the surrounding environmental scenario. Therefore, results from field testing are extremely context-dependent and should be interpreted with caution when being extrapolated to other environmental scenarios.

**Keywords:** honey bee; landscape context; foraging; climate; flower resources





## 1. Introduction

Honey bees provide a variety of products for humans [1]. They are mostly known for honey production but are also essential to provide pollination services both in natural ecosystems [2] and in agricultural areas, contributing to the pollination of approximately 35% of all crops [3,4]. Although human dependency on these services is increasing globally [5,6], the number of honey bee colonies is not increasing proportionally to match these needs [7,8].

The increased exposure to chemicals, diseases, and pests changes in beekeeping practices, and historical modifications to landscape composition, structure, and land use have created a cocktail of stressors that hinder the healthy development of colonies [9,10]. Intensive agriculture, in particular, has resulted in a high use of fertilizers and/or pesticides and landscape homogenization, resulting in there being fewer pollinator-friendly agricultural landscapes [11–13]. Furthermore, flower-rich semi-natural areas, including green

infrastructures such as field margins, hedgerows, and grasslands have been reduced or eliminated [14], leading to habitat fragmentation and loss [4]. As a result, honey bees experience a reduced or altered floral resource spatial and temporal availability in agricultural landscapes [4]). This may result in inadequate nutrition, affecting colony health and, ultimately, resulting in mortality [15,16], as colonies experiencing nutritional stress in poor resource landscapes are more prone to die in response to additional stressors [15,17]. Landscape composition has been suggested to be a key factor for honey bee colony development, with studies showing impacts on colony production [18–20], winter mortality [21], and development success for pollination services [22].

Additionally, landscapes are also influenced by climate through changes in the amount of flowering resources, phenology, and nectar production [23]. Climate also drives colonies' behaviour by shaping their interaction with the landscape through foraging activities [24], and by affecting how colonies respond to diseases and additional stressors [25].

Colony losses show high temporal and spatial variability [26,27], indicating a high influence of its past [28] and present environmental context. With this in mind, colony development and success should be context-dependent, with geographical differences affecting its behaviour and plasticity [29]. Nonetheless, field studies that evaluate colony development under different landscape scenarios are scarce [29,30], and spatio-temporally variable factors, including floral resource availability and climate, are usually poorly considered [31]. In most studies exploring the impact of landscape on honey bee colony development, satellite images are used to categorize general landscape composition (e.g., urban vs. agricultural [19]). However, these images usually have low resolution [32]; the qualitative categories resulting from these analyses are static and lack ecological relevance, as they do not quantify floral resources availability through time and space. More accurate resource evaluations are essential to explore the impacts of landscape composition on honey bee colony development.

The need to contextualize the development of colonies with their external environmental context is also relevant when considering the use of colony strength data for environmental risk assessment purposes [33,34]. If the background variability of honey bee colony size is to be considered for the definition of specific protection goals at the European level [34], there is an urgent need to document "normal" colony development in several representative landscapes.

The main goal of this study was to assess the impact of external environmental factors on honey bee colony development, by comparing three southern European landscapes located in the Iberian Peninsula with temporal and spatial variation in climatic conditions and resource availability. We hypothesize that there are key environmental variables leading colony development which could be used to evaluate the potential development of colonies under different landscapes.

Furthermore, we aimed to document the most important plant species used as nectar and pollen resources in these areas, as well as showing the trends of colony development in these landscapes. To accomplish these goals and to overcome the caveats of previous studies, we monitored colony development and key climatic variables, identified the main plant species used as nectar and pollen sources, and described temporal changes in floral resource availability to relate intrinsic colony parameters with external environmental variables.

## 2. Materials and Methods

### 2.1. Study Areas and Experimental Design

The study was carried out during three consecutive years in 10 km × 10 km study windows centered on three study apiaries that were installed at least a 1 km distance from other apiaries. The locations of these study areas are as follows: 2018 in Burgos (Spain), 2019 in Lousã (Portugal), and 2020 in Idanha-a-Nova (Portugal). The Burgos study window (42°16′51.0″ N 3°46′02.8″ W) was mainly composed of agricultural fields, being highly dominated by cereal crops, alfalfa, and sunflower. The study window in Lousã (40°02′53.6″ N 8°14′38.9″ W) was composed of broadleaf and deciduous forests, with



large areas of shrubland. Finally, the study window from Idanha-a-Nova (39°51′33.0″ N 7°09′49.7″ W) was an agricultural area, composed mostly of cattle pastures, with a few intensively managed permanent crops. Visual representation of the study windows can be found in the Supplementary Materials.

Each apiary consisted of five Apis mellifera iberiensis colonies in Langstroth hives provided by a professional beekeeper (with a complete track record of all the beekeeping practices applied). The colonies containing new queens were installed a year before monitoring; the colonies were managed according to local beekeeping practices and treated for varroa mites in the beginning (February–April) and at the end (July–September) of the season. Supplementary feeding in the form of sugar paste was provided at the beginning of the season in all apiaries due to the cold temperatures. All colonies were equipped with a Beeyard® scale that transmitted hourly weight and temperature data.

### 2.2. Resources Availability

Each study window was mapped using open-source GIS databases, following the use of CORINE land cover (level 3) to classify each polygon according to its land use category. Land cover of each polygon was later confirmed through field observations. To evaluate the cover (%), abundance (number of open flowers/$m^2$), and diversity of flowering plant species in each land use category, a minimum of two and a maximum of 20 sampling points, in different polygons from each category (within a 1.5 km radius from the apiary), were visited and visually assessed. At the Burgos landscape, this assessment was performed only for three months, whereas at Idanha and Lousã it was performed simultaneously with population assessment of colonies (once every 20 days for five months). For the Burgos landscape, a 1 $m^2$ square was used to estimate the overall polygon diversity and the abundance of each flowering plant species. Since both sampling frequency and the method to access polygon diversity and abundance in the Burgos landscape were found to be insufficient to capture the spatio-temporal resource availability, landscape data from Burgos was removed from the analysis and floral composition was compared only between Idanha and Lousã.

For the Idanha and Lousã landscapes, the polygon diversity and abundance of each flowering plant species was evaluated at the polygon level by using an estimate within a representative area. Furthermore, to evaluate resource availability at each sampling point, a resource score per $m^2$ was calculated. Each flowering plant species was classified according to a "Bee friendliness (BF) value" that was calculated based on each plant species beekeeping interest, nectar and pollen production, and overall honey bee visits [35]. The BF values of plant species not included in Alves da Silva et al. [35] were calculated as the average BF value of the genus or family. The BF value was multiplied by the flower abundance per $m^2$. To evaluate the total resource score of a certain land use category, the resource scores per $m^2$ of the polygons belonging to the same land use category were averaged and multiplied by the entire area of that category in the landscape.

Therefore, for each sampling date, each land use category was associated with a semi-quantitative value of resource offer (resource score), which was used to assess the temporal changes in resource availability in each study window.

### 2.3. Resources Collection

The amount and diversity of resources collected by honey bees was assessed through pollen traps and melissopalynological analyses for the three apiaries. Pollen traps (4.8 mm mesh) were installed for 24 h at the entrance of the colonies during each observation day. Pollen trapping periods longer than 24 h were avoided since bees can change their behaviour (e.g., increase foraging effort) if pollen is continuously removed [36]. After harvest, pollen samples were cleaned for debris, weighed (±0.01 g), and kept at −20 °C. Later, samples were dried at 40 °C for 48 h. For each sampling date and apiary, pollen samples from the five hives were homogenized and a common sample was sent to a certified laboratory for palynological analysis. In the laboratory, the pollen samples were completely

homogenized in water and pollen types were identified in a sub-sample of about 2.5 µg, mostly identified to plant family, genus, or species (based on the DIN-Norm-10760 [37]). The botanical composition of each sample was assessed by counting 500 pollen grains in the sample.

Melissopalynological analysis was performed using honey samples, collected from the harvested honey (pooled from all hives), to assess the most used plant species for nectar foraging. The percentage of pollen in the palynological analysis was calculated following Louveaux et al. [38], with a total of 1200 grains identified in each sample.

To examine the amount of collected pollen for each species and to avoid overrepresentation of smaller grains when considering only grain numbers, pollen diameter was used to calculate the weight of each pollen species, as follows [39]:

$$Mass_{i,j} = \frac{(n_i \ x \ d_i)_j}{\sum_i (n_i \ x \ d_i)_j} \times Mass_j,$$

where $i$ is the contribution of each pollen species to total mass (*Mass*, g) of collected pollen in a sample $j$, by weighing the species' occurrence frequency ($ni$, number of pollen grains of species $i$ averaged between the two sub-samples) by the pollen grain species-specific diameter ($di$).

The sum of pollen analyzed in each date throughout the season was calculated to have an overall representation of the most important flowering resources in the area. Pollen types representing <4% of the sample were considered minor and were reported in the other category.

### 2.4. Colony Development Parameters

From each apiary from the three landscapes, the five study colonies were subjected to health, strength, brood, and provision assessments every 20 days according to the protocol from Capela et al. [40]. Colony strength, i.e., the adult population, was assessed by weighing all the frames from the nest and honey suppers with and without bees, multiplying the weight difference by the weight of a single bee. Brood and beebread were assessed by analyzing high-quality images of each comb using the DeepBee® software [41], which classifies each alveolus by its content (eggs, larvae, pupae, nectar, honey, pollen, or "other"). Since nectar/honey cells are highly variable in weight, nectar/honey weight was calculated by subtracting the weight of wood frames, wax, brood, and beebread from the total frame weight. Disease prevalence was monitored by detecting clinical signs of the most common diseases. The levels of varroa mites were measured at each visit by counting the natural mite fall over a 48 h period, using the bottom board method [42], confirming low varroa levels for the duration of the study. During the field season, a log was kept of all colony and beekeeping activities (e.g., swarming events, supplementary feeding, and adding honey supper).

The hourly scale data were used to calculate the daily colony weight variation by subtracting the colony weight registered at 1 a.m. of each day with the value from the previous day at 1 a.m., since the colonies are not actively foraging at this period.

### 2.5. Climatic Variables

In each apiary, rainfall, wind speed and direction, solar radiation, temperature, and relative air humidity were registered every 15 min by a meteorological station (Watchdog 2900ET). Rainfall, solar radiation, and temperature (considered for all landscapes from 1 March to 21 September) were used to derive potential foraging minutes for each apiary. If rain was registered in the 15 min period, the potential foraging minutes were automatically set to zero. In the absence of rain, the potential foraging minutes per day (with a 15 min step) were calculated using the formula below, based on the work from Vicens & Bosch [43]:

$$r_s = 2261.9e^{-0.164t}$$

where $t$ is the external temperature (°C) and $r_s$ is the solar radiation threshold in w/m². When the real solar radiation was higher than the calculated threshold $r_s$, it was assumed that bees were able to carry out foraging activities. If the real solar radiation was lower than the calculated threshold, bees would not be able to perform foraging activities (represented by the gray area in Figure 1).

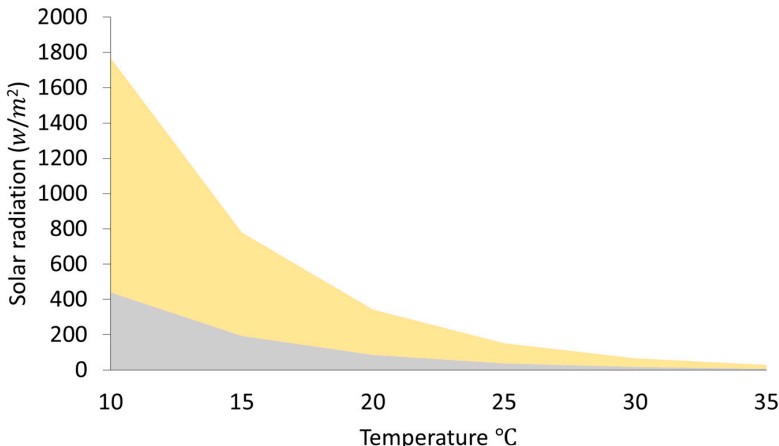

**Figure 1.** Solar radiation and temperature thresholds determining honey bee foraging activity. Foraging occurs above the grey area according to Vicens & Bosch (2000). The yellow area represents a theoretical threshold, in which the environmental variables are still sub-optimal allowing low foraging activity. Optimal environmental conditions allowing high foraging activity are assumed above the yellow area.

Furthermore, we developed a theoretical threshold of optimal and sub-optimal environmental conditions that lead to low foraging activity (sub-optimal environmental conditions; represented by the yellow area in Figure 1) or high foraging activity (optimal environmental conditions; represented by the area above the yellow area in Figure 1). Environmental conditions were considered optimal when the ratio between real solar radiation and the threshold was above 3 (shown in Figure 1 in the area above the yellow area).

*2.6. Data Analysis*

To assess which variables could affect overall colony weight dynamics, Linear Mixed Models (LMM) fitted by the Restricted Maximum Likelihood (REML) estimation method were used: daily available foraging minutes, minutes of low and high foraging activity, landscape resource score, and colony strength (number of adult bees) were considered explanatory variables and the colony daily weight variation (Cw) the response variable (Log X + 10). Colony ID was used as a random factor. For this analysis, only the productive season was considered, i.e., 120 days starting from when the weight of the colonies started to increase: from 25 April to 22 August in Lousã, and from 19 March to 16 July in Idanha. The daily foraging minutes and weight variation were calculated for each day using daily data, while the resources and colony strength were interpolated based on the assessments every 20 days. For interpolation, a FORECAST function (linear regression) was applied between two consecutive data points. Colonies that swarmed or lost the queen during the study were removed from the analysis. Therefore, for daily weight variation and colony strength, three colonies were considered for Lousã and five colonies for Idanha.

For this analysis, only the Lousã and Idanha datasets were used. Each dataset was analyzed separately, and explanatory variables were checked for collinearity by performing a data exploration; variables with high variance inflation factor values (VIF > 5; ref. [44]) were eliminated from the analyses. Testing models with different explanatory variables was performed by comparing Akaike values (AIC), and the models with the lowest AIC were selected. Analyses were performed using the Brodgar software (version 2.7.5).

## 3. Results

### 3.1. Resources Availability

The overall resource score for the two tested landscapes (Lousã and Idanha) showed a clear difference in the amount of resource availability and their temporal dynamics (Figure 2A,B). The Lousã landscape presented a high resource score, with a peak in May followed by a strong decline (Figure 2A). In contrast, the resource score in the Idanha landscape was low, presenting a peak in March followed by an almost linear decrease until July (Figure 2B).

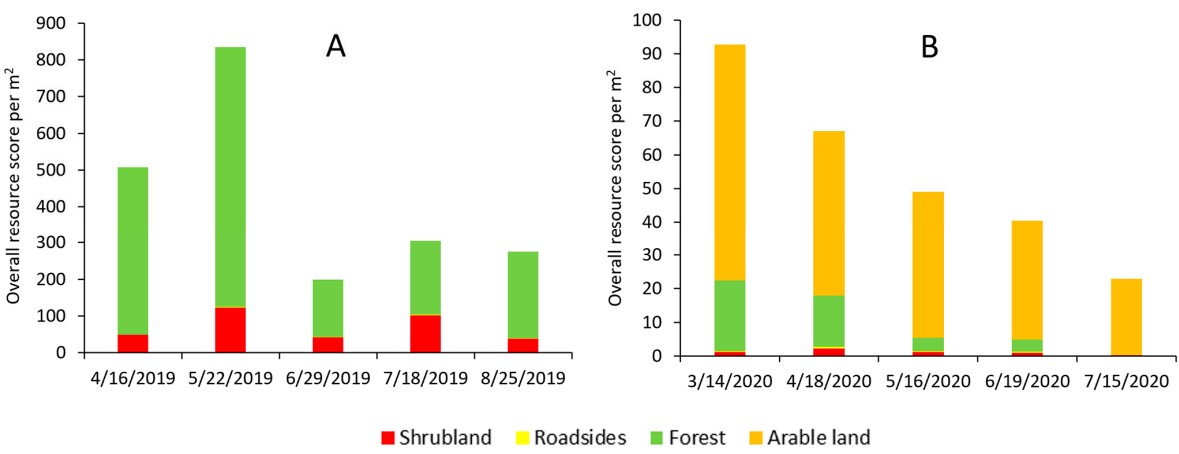

**Figure 2.** Resource score for the (**A**) Lousã and (**B**) Idanha landscapes (10 km × 10 km). Note the difference in scale of the resource score between landscapes.

### 3.2. Resources Collection

Considering the three landscapes, the colonies foraged for pollen from at least 37 different plant taxa throughout the season (Figure 3A). The genera Rubus, Eucalyptus, and Trifolium were collected in more than one landscape, while other genera such as Erica and Castanea were only found in Lousã.

In the Burgos landscape, sunflower (*Helianthus annus*), and clover (*Trifolium* sp.) were the most visited species for nectar collection. Clover was also the main nectar resource in the Idanha landscape, while in Lousã the bees foraged on heather (*Erica* sp.) and chestnut trees (*Castanea sativa*) for nectar collection (Figure 3).

### 3.3. Colony Development and External Environmental Variables

The colony development rate showed different trends considering the landscape (Figure 4). In all landscapes, the colony strength increased in Spring, with the peak achieved in April in Idanha, May in Lousã, and July in Burgos. The total number of brood cells also varied within colonies and landscapes. Contrary to colony strength, the number of brood cells in all apiaries had a similar peak at approximately 20,000 cells. Colonies in the Burgos landscape also produced more honey than the ones from Idanha or Lousã. Nectar/honey production lasted until 10 August in Burgos (2018), while colonies in Lousã (2019) and Idanha (2020) managed to increase nectar/honey storage until the middle of July and the middle of June, respectively. In all the scenarios, the number of beebread cells was kept at low levels while the colony was growing, and the beebread peak was usually achieved one to two months after the brood peak.

Overall (from 21 March to 21 September), the weather conditions allowed more daily foraging minutes in Idanha (2020), followed by Lousã (2019) and Burgos (2018; Figure 5). On the other hand, when considering only the productive season, Burgos (2018) climatic conditions granted more daily foraging minutes than Idanha (2020) and Lousã (2019; Figure 5).

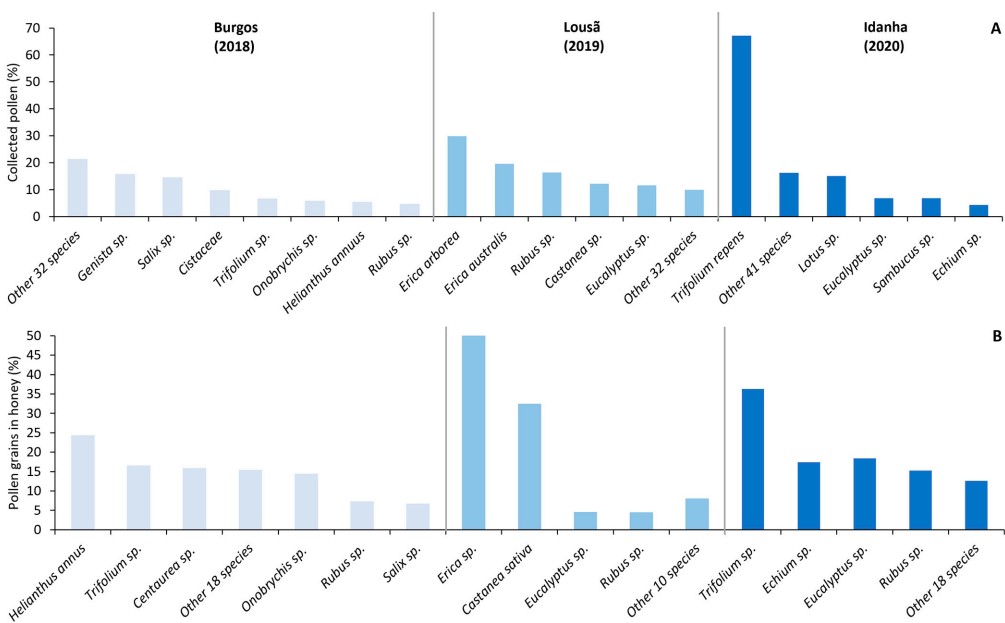

**Figure 3.** Relative percentage of (**A**) collected pollen and (**B**) pollen grains in honey for each landscape. Plant taxa with <4% presence were included in the "other" category.

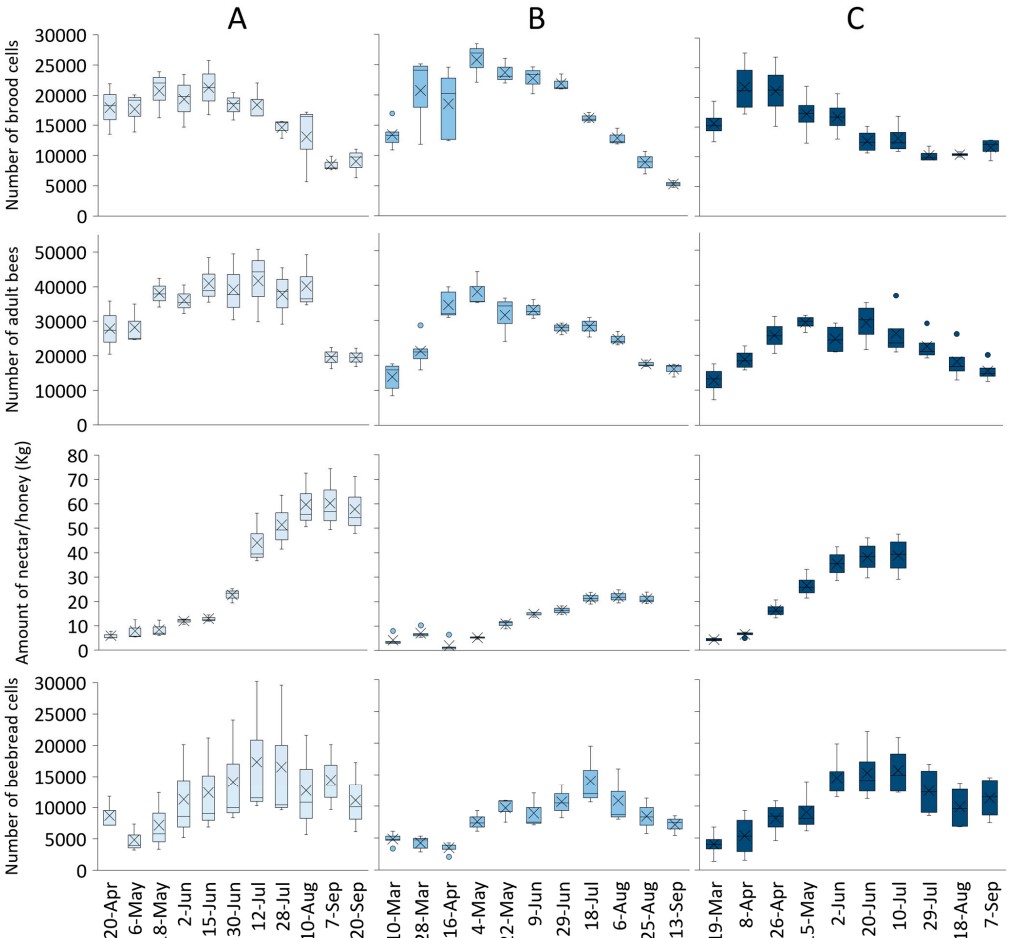

**Figure 4.** Colony development parameters (number of brood cells, number of adult bees, nectar/honey collection, and number of beebread cells) for (**A**) Burgos (2018), (**B**) Lousã (2019), and (**C**) Idanha (2020) landscapes.

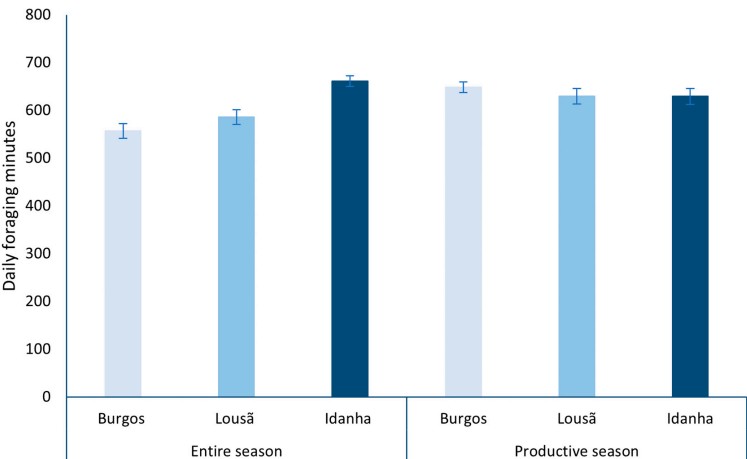

**Figure 5.** Daily mean of available foraging minutes in each landscape considering the entire season and only the productive season.

In Lousã, the variation in colony weight (Cw) was significantly explained by colony strength (positive relationship; $p = 0.0001$; Figure 6a) and the total amount of foraging time (positive relationship; $p < 0.0001$; Figure 6a) and by the amount of sub-optimal foraging minutes (negative relationship; $p < 0.0001$; Figure 6a). However, although resource availability was positively related to colony weight during the season, it did not explain a significant proportion of its variation. In contrast, resource availability at Idanha was the only variable that significantly explained the variation in colony weight (positive relationship; $p < 0.0001$; Figure 6b), while foraging times and colony strength were not significant predictors.

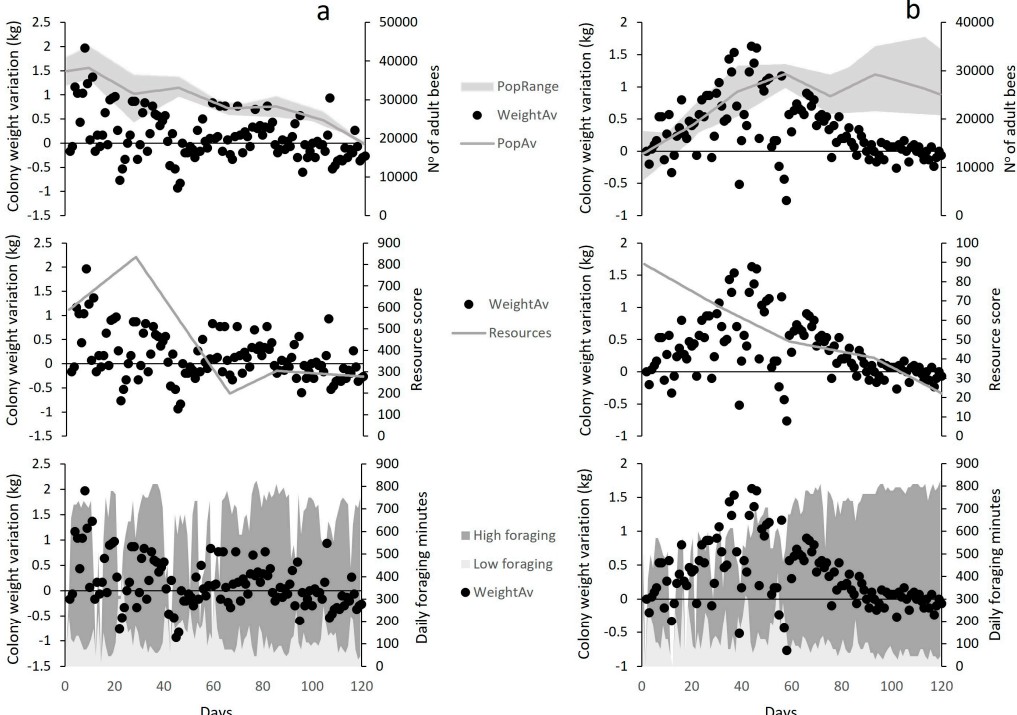

**Figure 6.** Visual representation of explanatory variables behaviour in relation to colony weight variation (WeightAv) at (**a**) Lousã and (**b**) Idanha landscapes. PopRange = Colonies strength range; WeightAv = Weight Variation Average; PopAv = Colony Strength aAverage; resources = landscape resource score; high foraging = mean daily number of optimal (high) foraging minutes; low foraging = mean daily number of sub-optimal (low) foraging minutes.

## 4. Discussion

Honey bee colonies showed different patterns of colony development within and between the three landscapes tested, suggesting that intrinsic internal mechanisms drive colony development while being influenced by external environmental variables. The resource score applied in this study allowed, for the first time, a semi-quantitative measurement of floral resource availability. In landscapes with low levels of floral resources (i.e., Idanha) flower availability appears to be a limiting factor for colony development. On the other hand, in landscapes with high resource levels (i.e., Lousã) climatic variables limiting seasonal foraging time (foraging minutes) and colony strength (number of foragers) seem to be the major drivers of colony weight variation.

### 4.1. Resource Availability in the Landscapes

In all tested scenarios, floral resource availability was expected to be spatio-temporally dynamic. This expectation was confirmed by field observations of flower resources in each land use category, and the obtained resource score for Lousã and Idanha showed significantly different resource availability, later linked with colony development. These results thus support the importance of accurately quantifying floral resources to assess their impact on colony development. Data from flower resource production is still scarce and, when available, is somewhat specific for certain climatic regions [45,46]. Consequently, we still lack information about important honey bee production areas such as southern European landscapes. Here, we provide for the first time a semi-quantitative approach for these regions based on field assessments and on the "Bee Friendly value" from Alves da Silva et al. [35]. Field assessments provide real estimates of floral availability and diversity in the landscape, while the BF value enables us to score plant species based on their nectar and pollen production and frequency of visits by honey bees. Although this approach does not directly measure nectar and pollen production, considering the lack of bibliographic information for nectar and pollen production and the highly time-consuming task of obtaining them in the field, the approach used here is a good proxy for resource availability.

### 4.2. Resource Collection

The composition of floral resources collected by honey bees also varied between landscapes, most likely reflecting the documented differences between the study landscapes. In the Burgos landscape, *Genista* spp. and *Salix* spp. were the most visited genera for pollen collection, while the sunflower was the main resource for nectar collection. Despite the high presence of arable land (66% used for cereal crops, 9% sunflower, 3% alfalfa), the colonies relied on the use of trees and shrubs to collect pollen for their colony development. A reliance on these types of resources has already been shown in several studies [47,48]. Furthermore, bees also relied on wildflowers from the genera *Trifolium* spp., *Centaurea* spp., and *Onobrychis* spp., and from the Cistaceae family before the sunflower bloom. The use of these resources shows that bees are able to find small patches of natural habitats in an agricultural landscape dominated by crop fields, which is essential for their development [15,39].

In Idanha, the landscape was also dominated by arable land, although it was mostly permanent pastures. These include grasslands with grass and other herbaceous plants (e.g., *Trifolium* spp. and *Echium* spp.), which were evidently used by bees for pollen and nectar collection. A previous study conducted in the same region identified other relevant species (besides *Echium* spp.) for honey production (i.e., *Erica* spp., *Lavandula* spp., and *Campanula* spp. [49]). These were also observed in the study area, though with a lower percentage of coverage, evidencing the local influence of the plant communities.

In Lousã, the landscape was dominated by forest and shrubland, with only a small proportion of arable land. Here, the bees collected pollen and nectar from heather (*Erica* spp.), chestnut (*Castanea sativa*), and brambles (*Rubus* sp.). These species are commonly found in forested areas and used by honey bees in Portugal [50,51]. Moreover, in both Portuguese landscapes (i.e., Lousã and Idanha), bees used eucalyptus spp. flowers [52]. This plant genus is widely distributed in Portugal, with a high proportion of forested area being

covered by eucalyptus monoculture [53]. These areas are used by beekeepers mainly in the littoral area to produce honey during the winter.

### 4.3. Colony Development

Regional differences in landscape composition and weather conditions contributed to a differential development of the colonies. In Burgos, colonies were able to maintain a large brood size during a prolonged period (approximately 20,000 alveoli for three months), leading to high population levels until August. Interestingly, the population peak was achieved right before the sunflower blooming period, ensuring that many forager bees collected resources from this nectar-rich crop, simultaneously providing the highly needed pollination services for this crop. In this scenario, both beekeepers and farmers benefit from the presence of non-arable polygons in the landscape, which are necessary to support the growth of colonies to reach their peak strength prior to sunflower blooming. This scenario, (reaching their population peak before the sunflower boom) has already been documented in one of the most comprehensive studies to measure detailed honey bee colony dynamic data under real beekeeping management conditions [30]. In Idanha and Lousã, the maximum colony sizes were similar, but showed a slight chronological desynchronization. These levels of brood (from 20 to 28 thousand alveoli) have also been measured in other experiments during summer months [29]. Nonetheless, as in this study, these levels are highly variable and the authors concluded that the year of the test significantly affected brood production [29], evidencing an influence of the climatic variables. Indeed, this desynchronization between the study sites can be explained by the temporal pattern of brood rearing, as brood production is closely related to diet and external temperature [54,55]. Remarkably, in all apiaries, the levels of beebread peaked later in the season, after the peak of brood levels. As the brood pheromone [56,57] is one of the main drivers of colony mechanisms for pollen foraging, it may be expected that a reduction in brood production may induce reduced pollen collection. Nonetheless, in late season, when the nectar/sugar availability is reduced, the forager bees change their foraging efforts towards pollen [58], leading to the accumulation of beebread in the colony, which becomes essential for winter bee rearing when pollen availability is scarce.

The colonies installed in agricultural areas (i.e., Burgos and Idanha) had a higher honey yield than in the forested area (i.e., Lousã). The mean daily available foraging minutes, as well as the achieved population levels, can partially explain these differences. Nonetheless, we hypothesized that the landscape resources availability would also play a fundamental role on colony production levels. From the Idanha analysis, it was possible to conclude that the available resources play a significant role in the daily weight variation. On the other hand, in the Lousã analysis, the landscape resources did not explain a significant variation in colony daily weight change. In this case, we hypothesize that environmental conditions for foraging and colony strength are more limiting than the availability of floral resources. Furthermore, when the environmental conditions are sub-optimal for foraging activity (i.e., low foraging), the colony has a poorer performance.

The model used to derive foraging time and the exploration of sub-optimal vs. optimal climate conditions allowed us to calculate the available foraging minutes. However, it does not provide a continuous prediction of how a change in climatic variables translates to a change in foraging activity (the possible number of bees leaving and entering the colony), and it does not consider the upper limits of solar radiation and temperature. Thus, we believe this method should be further explored, as it can provide ecologically relevant endpoints, since climate is one of the main drivers of bee foraging behaviour [24]. Understanding the local dynamics between environmental variables and colonies' status can help beekeepers to apply better beekeeping practices to reduce colonies mortality [59,60]. This is particularly relevant considering that beekeeping practices have been appointed as a major cause influencing colonies' development and success [61,62].

Additionally, although pests and diseases were controlled through visual assessment, to exclusively measure the influence of the landscape context on the colonies, their development may still be affected by diseases despite symptoms not being visible [63]. Furthermore, the genetic variability of colonies may also influence their development [64]. In the current study, we tried to reduce the influence of genetic composition by using experimental colonies from the same region. Hence, we believe that colony development patterns were strongly linked to environmental conditions in the landscape where the apiaries were installed [20,29,65].

### 4.4. Final Remarks

The correlation between landscape composition and honey bee colony development is not new and, despite the advances made here, still needs to be further explored [31]. However, quantifying floral resources at the landscape-level still presents major limitations. Most available land cover products lack sufficient local accuracy to correctly describe and measure the temporal and spatial shifts of floral resources, as they only provide land use categories [32]. There is still the need to associate flowering composition with land use categories. On one hand, this might be easily solved in areas mainly composed of arable land by relying on national databases, which have information on farmers' activities, or by using satellite images paired with other techniques (e.g., NDVI) that allow crops identification [66,67]. On the other hand, in landscapes partially composed of several, usually small, patches used for local farming, and/or in which the national spatial databases are incomplete, this task becomes more challenging. Furthermore, flower abundance from field assessments cannot be used in different areas of the same land use type if the field topology and climatic characteristics differ. Only by integrating habitat-specific species composition and phenology with climate variables [67] and nectar and pollen production (from which there is still a scarcity of data), would it be possible to create landscapes with a continuous (e.g., daily) spatial and temporal quantification of available resources for honey bees to mechanistically link the resources collection with their availability.

Some of the plant species present in the study landscapes (i.e., *Trifolium* sp., *Salix* sp., and *Helianthus* sp.) are commonly used by honey bees at the European scale [68]. Nevertheless, this study documents the need for regional scale studies with, for example, the use of sentinel honey bee colonies, which can be a good tool to evaluate resources' availability and even potential exposure to plant protection products. This approach was proposed by the EFSA scientific committee [33] to gather data on pesticide exposure and effects for a pre- and post-approval environmental risk assessment system and is currently being applied in some European projects (e.g., INSIGNIA-EU project). Furthermore, the use of sentinel colonies could overcome the caveats associated with pollen and nectar resources availability assessments at the landscape scale, as they are extremely laborious and rely on an anthropomorphic evaluation. The use of sentinel hives could therefore complement the construction/modelling of detailed landscapes at a regional scale, which are extremely important for the development of predictive models to overcome the challenges of field testing.

## 5. Conclusions

In this study, colonies changed and adapted their behaviour in response to external variables, leading to different development rates, showing that results from field testing are extremely context-dependent; intrinsic internal mechanisms drive colony development, while being influenced by external environmental variables. Therefore, the use of field testing in environmental risk assessment is valuable for specific contexts, but extrapolations should be made with caution.

**Supplementary Materials:** The following supporting information can be downloaded at: https://www.mdpi.com/article/10.3390/d15121188/s1, Figure S1: Visual representation of landscapes in Lousã, Idanha and Burgos.

**Author Contributions:** Conceptualization, N.C. and J.P.S.; methodology, N.C., A.S., S.C. and J.A.; formal analysis, J.P.S. and A.A.d.S.; investigation, N.C., S.S. and S.L.; data curation, N.C. and Y.L.D.; writing—original draft preparation, N.C. and A.S.; writing—review and editing, all authors; supervision, J.P.S.; funding acquisition, Y.L.D., D.C.d.G. and J.P.S. All authors have read and agreed to the published version of the manuscript.

**Funding:** This work was funded by the EU INTERREG-SUDOE POLL-OLE-GI project (SOE1/P5/E0129), by the European Food Safety Authority (Ref: OC/EFSA/SCER/2017/02) and by the H2020 B-GOOD project (Grant Agreement 817622).

**Institutional Review Board Statement:** Not applicable.

**Data Availability Statement:** The data presented in this study are available on request from the corresponding author. The data are not publicly available due to privacy.

**Acknowledgments:** We would like to thank Pedro Martins da Silva for his help on data analysis, and Mari Gigauri, Ruben Mina, Daniela Mendes, Lucie Mota, and Caio Domingues for their help during field work on colonies and landscape assessments. We would also like to give a special thanks to Henrique Azevedo-Pereira for his help when setting the initial experiments.

**Conflicts of Interest:** The authors declare no conflict of interest.

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
