# Peer review of "Exploring the External Environmental Drivers of Honey Bee Colony Development"

_diversity, doi:10.3390/d15121188_

Round 1
Reviewer 1 Report
Comments and Suggestions for Authors
Please find the comments.
Abstract:
·Please add a brief conclusion at the end of abstract.
Introduction:
·Line 32: Add a recent reference. Authors can utilize http://dx.doi.org/10.1016/j.sjbs.2016.12.009
·Line 65: "including floral resource availability and climate“ including floral resource availability, subspecies, and climate. Authors can utilize https://doi.org/10.1016/j.sjbs.2017.10.009
· Add a paragraph about the major nectar and pollen resources in the study areas.
Materials and Methods Pollen grains in honey
· Authors did not mention how determine the pollen grains in honey.
· In the statistical analysis section, The type of ANOVA used in the experiment should be mentioned.
Results
· At the vertical pars In Fig. 3, write the numbers without %.
Discussion
·Authors can compare their plant with other plants in Egypt. They could utilize https://doi.org/10.1016/j.sjbs.2017.12.010
· Authors can compare their pollen, brood, and worker numbers with other in Saudi Arabia. They could utilize https://doi.org/10.1016/j.sjbs.2017.10.009
· Please add a conclusion.
Comments on the Quality of English LanguageModerate editing of English language required.
Author Response
Dear reviwer, our replies are in bold. Thank you for your time and input in the manuscript.
Abstract:
- Please add a brief conclusion at the end of abstract.
A conclusion was added and it was indeed missing.
Introduction:
- Line 32: Add a recent reference. Authors can utilize http://dx.doi.org/10.1016/j.sjbs.2016.12.009
- This study is not relevant for that paragraph.
- Line 65: "including floral resource availability and climate“ including floral resource availability, subspecies,and climate. Authors can utilize https://doi.org/10.1016/j.sjbs.2017.10.009
- This study is not relevant for that paragraph since we really want to focus on climate and floral resources. We know that subspecies influence the outcome of the study since they will show respond differently to the external variables. Nonetheless, we assume that studies, like ours, use locally adapted colonies.
- Add a paragraph about the major nectar and pollen resources in the study areas.
- That was one of the results of the study. We should not add results in the introduction or else, the reader might be confused.
Materials and Methods Pollen grains in honey
- Authors did not mention how determine the pollen grains in honey.
- It was added to the manuscript the methods for both pollen and honey analysis. Before, methods were not added since this was externally performed by two certified laboratories.
- In the statistical analysis section, The type of ANOVA used in the experiment should be mentioned.
- It was not used any kind of ANOVA in this study. The data was analyzed using Linear Mixed Models.
Results
- At the vertical pars In Fig. 3, write the numbers without %.
- Done. Thank you for the suggestion.
Discussion
- Authors can compare their plant with other plants in Egypt. They could utilize https://doi.org/10.1016/j.sjbs.2017.12.010
- In this study we searched for local studies in the same regions to show that the selected areas are somehow representative of these regions. So, we believe it does not make sense to add studies from areas with totally different climate or flower availability.
- Authors can compare their pollen, brood, and worker numbers with other in Saudi Arabia. They could utilize https://doi.org/10.1016/j.sjbs.2017.10.009
- We agree that we need to compare brood and population numbers with other studies. Thank you for the suggestion.
- Please add a conclusion.
The conclusion was in the last paragraph of the final remarks. Now it has its own section. Thank you for the suggestion.
Reviewer 2 Report
Comments and Suggestions for Authors
This paper investigates the relationship between resource availability, honey bee foraging behavior, and colony strength across different landscapes using a novel approach that combines field observations, melissopalynological analyses, and colony development parameters. The study's implementation of a "Bee friendliness (BF) value" for different flowering plant species, based on their beekeeping interest and nectar and pollen production (Lines 122-124), is an innovative method that could potentially be adopted in future beekeeping research. The research methodology is well documented and executed, such as the calculation of a resource score per m2 (Lines 121-122) and the use of pollen traps for resource collection (Lines 134-136). The study also carefully considers a range of variables that could influence the results, including climatic variables (Lines 171-173) and colony development parameters (Lines 152-154). The findings could have important implications for understanding how environmental conditions and resource availability impact honey bee behavior and health, which is of significant relevance to the field of beekeeping and environmental biology.
Specific comments:
Abstract:
The abstract provides a concise summary of the study. However, it would be helpful to include more specific information about the findings and their implications. Additionally, consider including the sample size and key methods used in the study.
Introduction:
The introduction provides a clear overview of the importance of honey bees in agricultural landscapes and the challenges they face. Consider expanding on the specific stressors that impact colony development and providing more context for the previous studies mentioned.
Materials and Methods:
The section on study areas and experimental design is comprehensive and provides essential information. However, it would be beneficial to include details about the sample size and any statistical analyses performed. Additionally, consider providing more information about the specific data collection methods used, such as how colony development and climatic variables were monitored.
Results:
The results section should present the key findings of the study. Include specific data and statistical analyses where appropriate. Consider organizing the results into subsections to improve readability and clarity.
Discussion:
The discussion should interpret the results and relate them to the research objectives. Discuss the implications of the findings in the context of existing literature. Consider addressing any limitations of the study and suggesting avenues for future research.
Conclusion:
You should provide a conclusion section that summarizes your findings and provides the impact of these findings on the field.
In general Weaknesses:
1. The paper lacks a clear presentation of the results and the statistical analyses used to derive them. For example, it is not evident from the text how the authors determined the significance of the relationships between variables such as colony weight, foraging minutes, and resource score (Lines 265-267 and 269-270).
2. The discussion and interpretation of the findings are missing, making it difficult for readers to understand the implications of the results. For instance, the implications of the different trends in colony development across the three landscapes (Figure 4) are not discussed.
3. The paper does not provide a clear research question or hypothesis, making it difficult to discern the aim of the study.
4. The introduction and background information provided are insufficient to contextualize the study and its relevance.
Comments on the Quality of English LanguageMinor editing of English language required
Author Response
Dear reviwer, thank you for your time and valuable input in the manuscript. Our reply is in bold.
This paper investigates the relationship between resource availability, honey bee foraging behavior, and colony strength across different landscapes using a novel approach that combines field observations, melissopalynological analyses, and colony development parameters. The study's implementation of a "Bee friendliness (BF) value" for different flowering plant species, based on their beekeeping interest and nectar and pollen production (Lines 122-124), is an innovative method that could potentially be adopted in future beekeeping research. The research methodology is well documented and executed, such as the calculation of a resource score per m2 (Lines 121-122) and the use of pollen traps for resource collection (Lines 134-136). The study also carefully considers a range of variables that could influence the results, including climatic variables (Lines 171-173) and colony development parameters (Lines 152-154). The findings could have important implications for understanding how environmental conditions and resource availability impact honey bee behavior and health, which is of significant relevance to the field of beekeeping and environmental biology.
Specific comments:
Abstract:
The abstract provides a concise summary of the study. However, it would be helpful to include more specific information about the findings and their implications. Additionally, consider including the sample size and key methods used in the study.
Thank you for the suggestion. Your study description was even used as inspiration to improve the abstract.
Introduction:
The introduction provides a clear overview of the importance of honey bees in agricultural landscapes and the challenges they face. Consider expanding on the specific stressors that impact colony development and providing more context for the previous studies mentioned.
We have decided not to extend the specific stressors section because this was not the focus of the study. Instead, we decided to keep the focus on the stressors that we are studying. We know that there are several reviews on honey bee multiple stressors, but it was not relevant for this study to explore all of those stressors in the introduction. Hope you can understand.
Materials and Methods:
The section on study areas and experimental design is comprehensive and provides essential information. However, it would be beneficial to include details about the sample size and any statistical analyses performed. Additionally, consider providing more information about the specific data collection methods used, such as how colony development and climatic variables were monitored.
The statistical analysis that were performed are in the methods section. As an example, we did not make any statistical analysis in the colony development data alone (to compare between landscapes) because the goal is to showcase trends and not to simple say that the areas are different. On the other hand, for the main goal of the study, to know which external environmental variables influence the colonies development, statistical analysis were thoroughly described.
Regarding the methodology, we decided to present a summary of the techniques (but also including some details) with all the references from other studies that used these protocols. We felt that there was no need to make an extensive description on the colonies measurements because there is already a protocol published with it (the reference is in the text).
The climatic variables were measured using a weather station and it is well described in the “climatic variables” section how the variables that we extracted, their periodicity and how we used those data in our study.
Results:
The results section should present the key findings of the study. Include specific data and statistical analyses where appropriate. Consider organizing the results into subsections to improve readability and clarity.
The results section is already organized into subsections (Resources availability, Resources collection, and Colony development and external environmental variables) as the material and methods.
Discussion:
The discussion should interpret the results and relate them to the research objectives. Discuss the implications of the findings in the context of existing literature. Consider addressing any limitations of the study and suggesting avenues for future research.
Thank you for your suggestions. We have upgraded the discussion to add more studies that relate to our research.
Conclusion:
You should provide a conclusion section that summarizes your findings and provides the impact of these findings on the field.
The main conclusion of the study was in the last paragraph of the final remarks. It was changed to make it clear for the reader.
In general Weaknesses:
- The paper lacks a clear presentation of the results and the statistical analyses used to derive them. For example, it is not evident from the text how the authors determined the significance of the relationships between variables such as colony weight, foraging minutes, and resource score (Lines 265-267 and 269-270).
It is explained, in the methods section, how the statistical analysis was performed. Maybe, the graphs in the supplementary material can help the reader to develop a clear picture of how these variables interact (in a positive or negative way) with each other. Therefore, that section was upgraded to have more information in the text and the graphs were added in the main document.
- The discussion and interpretation of the findings are missing, making it difficult for readers to understand the implications of the results. For instance, the implications of the different trends in colony development across the three landscapes (Figure 4) are not discussed.
This (colony development data) was indeed a flaw in our discussion and it was now upgraded.
- The paper does not provide a clear research question or hypothesis, making it difficult to discern the aim of the study.
We believe that the research question is presented as the main goal. Nonetheless, we agree that the hypothesis were not stated in the introduction and are now included.
- The introduction and background information provided are insufficient to contextualize the study and its relevance.
To provide extra background information, another paragraph was added. The new paragraph, about documenting the normal colony development was something discussed in the discussion, but it was never in the introduction. Thank you for the suggestion.
Reviewer 3 Report
Comments and Suggestions for Authors
The paper is well written and approaches a fascinating subject, which is also, at the same time, challenging to study considering the difficulties in field trials.
Despite the effort, I am wary of being able to draw any solid conclusions from comparing the different sites over time, as measurements have been made for one year at each site.
As is known, the weather conditions and the effect they may have on bees are different, so comparison between regions cannot be considered safe. Possibly different results would have been obtained if the three apiaries had been installed in the same year and compared in the different study areas.
Also, I think it is essential to provide information about the presence of other bee colonies at the same time of the experiment. If it is a common practice to transport many bee-colonies to the selected areas (e.g., sunflowers), the presence of many bees in the tested plants could affect the research results.
A minor comment concerns the statement in lines 384-385: “In the current study, we assume a low influence of genetic composition since the queens of the experimental colonies were from the same region.” The resemblance between queens and the resulting behaviour of bees is not established by their rearing in the same region. Even when sister – queens are used, there are differences in the genetic material, giving significant variability among bee colonies. The main reason for this is that queens will mate with many drones from different colonies.
In the attached PDF file, I give comments to correct some (very few) written mistakes.

The paper is well written. Minor editing of English language required
Author Response
Dear reviewer, thank you for your time and input in the manuscript. Our replies are in bold.
The paper is well written and approaches a fascinating subject, which is also, at the same time, challenging to study considering the difficulties in field trials.
Despite the effort, I am wary of being able to draw any solid conclusions from comparing the different sites over time, as measurements have been made for one year at each site.
As is known, the weather conditions and the effect they may have on bees are different, so comparison between regions cannot be considered safe. Possibly different results would have been obtained if the three apiaries had been installed in the same year and compared in the different study areas.
Dear editor. Thank you for your input. We also know that the field trials bring us a lot of variability and it is hard to compare between sites. But that was just the first part of the story – show how colonies develop. Having different sites allowed us to document the “normal” colony development in these areas. And, even if we did the experiments in the same year, the climate in each area would be different. So, the climatic variables would always be different between experimental sites. Bearing this in mind, the first goal was just to show some trends, lows and highs, of colonies development for these different regions and the climatic variables would always be different.
Then, comes the second part of the story. Based on literature research we know that the external variables (as climate and flower availability) are some of the main drivers of this development. But most of the literature was based on theoretical evidence because it is hard to measure some of these variables (ex. resources availability).
So, our main goal was to carefully measure the external variables and find variables that could explain the colony development in the different regions. In the beginning we hypothesized that the main drivers were the same for all the landscapes. To our wonder, it seems that this is not true. The external environmental context have a strong influence but that influence is not the same in different landscapes.
Also, I think it is essential to provide information about the presence of other bee colonies at the same time of the experiment. If it is a common practice to transport many bee-colonies to the selected areas (e.g., sunflowers), the presence of many bees in the tested plants could affect the research results.
Thanks for your comment and we totally agree with the statement. The experiments were done in areas with low presence of other beekeepers. The only apiary that we were concerned about was the one in Burgos, because of the sunflower’s presence. Thankfully, since the farmers haven’t felt the need to hire pollination services, we managed to go through the season without the presence of other colonies. This information regarding the distance to other apiaries was now added into the manuscript.
A minor comment concerns the statement in lines 384-385: “In the current study, we assume a low influence of genetic composition since the queens of the experimental colonies were from the same region.” The resemblance between queens and the resulting behaviour of bees is not established by their rearing in the same region. Even when sister – queens are used, there are differences in the genetic material, giving significant variability among bee colonies. The main reason for this is that queens will mate with many drones from different colonies.
It is true that they are affected by their matting, even if it is done at the “local” level. In this case, we were trying to state the flaws of the study and finding some justifications that could soften those flaws, since most of them were out of our control. I believe that now, the justification is more “honest” than before.
In the attached PDF file, I give comments to correct some (very few) written mistakes.
We really appreciate this PDF and the English corrections.
Round 2
Reviewer 1 Report
Comments and Suggestions for Authors
Introduction:
·Line 32: Add a recent reference.
·Line 65: "including floral resource availability and climate“ including floral resource availability, subspecies, and climate.
· Add a paragraph about the major nectar and pollen resources in the study areas.
Materials and Methods
· In the statistical analysis section, The type of ANOVA used in the experiment should be mentioned. How the authors compare between the means?
Discussion
·Authors should compare their plant with other plants in area.
· Authors should compare their pollen, brood, and worker numbers with other in world.
Comments on the Quality of English Language
Moderate editing of English language required
Author Response
- Line 32: Add a recent reference.
- - Two new references were added.
- Line 65: "including floral resource availability and climate“ including floral resource availability, subspecies,and climate.
- - We did not add subspecies because it is not considered a spatiotemporal variable.
- Add a paragraph about the major nectar and pollen resources in the study areas.
- - This is one of the goals of the study. Therefore, it does not make sense to describe our results in the introduction.
Materials and Methods
- In the statistical analysis section, The type of ANOVA used in the experiment should be mentioned. How the authors compare between the means?
- - We used Linear Mixed Models to compare between the means.
Discussion
- Authors should compare their plant with other plants in area.
- - Currently, we are comparing with several other studies from the same area in order to show that these areas are representative of these regions.
- Authors should compare their pollen, brood, and worker numbers with other in world.
- - In this version we have already included other studies that have evaluated colonies development, and compared our results with theirs, based on the comment that was made in the last revision.
Reviewer 3 Report
Comments and Suggestions for Authors
I have written the manuscript carefully under the authors’ view, considering their comments regarding how they have approached this fascinating subject.
The climate and other parameters would differ even if the apiaries were installed in the same year at different sites. But, even this way, it would give the most solid results.
In my opinion, the only way these findings can be used is to avoid making any comparisons between them. Especially in the following points:
Lines 248-253: Try to avoid the comparison of two landscapes. Especially the comparison of the month where the peak occurred. It is essential to refer to the month for each case, but only as data and not as a parameter to compare.
Line 267 – 268: In the figure 3, you must mention the place and the year. A good example is in figure 4 (lines 286-287), where the difference in year experiment is more obvious.
Line 274: Mention the year next to the landscape, not only the month, because it gives the information that you compare the same year’s results.
Line 277: Also mention again the year. You compare the production of honey using dates from different years.
Line 278-280: Always refer to specific years of the compared parameters.
Line 288-289: I'm afraid I have to disagree with comparing different years’ data and their presentation with specific dates. Every year has other characteristics, and estimating specific periods is quite problematic. I would suggest removing this sentence.
The way that the authors can present their data and find me agree is in lines of manuscript 296-305. Individual each landscape and avoid to compare.
Author Response
I have written the manuscript carefully under the authors’ view, considering their comments regarding how they have approached this fascinating subject.
The climate and other parameters would differ even if the apiaries were installed in the same year at different sites. But, even this way, it would give the most solid results.
In my opinion, the only way these findings can be used is to avoid making any comparisons between them. Especially in the following points:
- We totally understand your point of view. To clarify that this is not just a compasison between landscapes but that climate variables are always playing a role on the output that we got.
Lines 248-253: Try to avoid the comparison of two landscapes. Especially the comparison of the month where the peak occurred. It is essential to refer to the month for each case, but only as data and not as a parameter to compare.
- Changes were made to just state if it is high or low, without a clear comparision between the landscapes.
Line 267 – 268: In the figure 3, you must mention the place and the year. A good example is in figure 4 (lines 286-287), where the difference in year experiment is more obvious.
- In here, the pollen data is cummulative. It is the sum of all year's sampling. Therefore, we believe it is strongly connected with the landscape and poorly connected with the year. But, we can also understand that the climate patterns of a certain year can contribute to longer or shorter flowering periods of some plants. Having this in mind, the year was added to the graph, but we believe this data is strong enough to have an overview of the most important plant species in these landscapes, independent of the climate and year.
Line 274: Mention the year next to the landscape, not only the month, because it gives the information that you compare the same year’s results.
- The suggestion was added.
Line 277: Also mention again the year. You compare the production of honey using dates from different years.
- The suggestion was added.
Line 278-280: Always refer to specific years of the compared parameters.
- The suggestion was added.
Line 288-289: I'm afraid I have to disagree with comparing different years’ data and their presentation with specific dates. Every year has other characteristics, and estimating specific periods is quite problematic. I would suggest removing this sentence.
- In this case, we are showing the climatic conditions of a specific landscape for a specific year of the experiment and how those conditions can lead to more or less foraging minutes. This data was then used as a explantory variable for the weight data. We don't want to state that the weather patterns are always the same in these landscapes. We are using that years weather as an explanatory variable of colonies development.
The way that the authors can present their data and find me agree is in lines of manuscript 296-305. Individual each landscape and avoid to compare.
- We understand that the years are different but, once again, even if the year was the same, what really matters is that we used the climatic data to explore what happens in each landscape. We could discuss all of these results without comparing landscapes and refering to each of the landscapes as a case study. Nontheless, we believe that some comparisons need to be made, when it comes to general patterns, to enhance the discussion.